# Suppression of Platelet-Derived Growth Factor Receptor-Alpha Overcomes Resistance to Trastuzumab through STAT3-Dependent IL-6 Reduction in HER2-Positive Breast Cancer Cells

**DOI:** 10.3390/biomedicines11030675

**Published:** 2023-02-23

**Authors:** Sangmin Kim, Hyungjoo Kim, Yisun Jeong, Daeun You, Sun Young Yoon, Eunji Lo, Seok Jin Nam, Jeong Eon Lee, Seok Won Kim

**Affiliations:** 1Department of Breast Cancer Center, Samsung Medical Center, 81 Irwon-Ro, Gangnam-gu, Seoul 06351, Republic of Korea; 2Department of Surgery, Samsung Medical Center, Sungkyunkwan University School of Medicine, 81 Irwon-Ro, Gangnam-gu, Seoul 06351, Republic of Korea; 3Department of Health Sciences and Technology, SAIHST, Sungkyunkwan University, 81 Irwon-Ro, Gangnam-gu, Seoul 06351, Republic of Korea

**Keywords:** PDGFRA, IL-6, trastuzumab resistance, HER2+ breast cancer

## Abstract

Platelet-derived growth factor receptor (PDGFR) plays an essential role in the proliferation and invasion of malignant cancer cells. However, the functional role of PDGFR alpha (PDGFRA) in HER2-positive (HER2+) breast cancer has not been fully clarified yet. Thus, the objective of this study was to investigate the clinical significance of PDGFRA and the therapeutic potential of PDGFR inhibitors as part of an effort to overcome trastuzumab (TRZ) resistance. Aberrant PDGFRA expression is closely associated with decreased survival in HER2+ breast cancers. Therefore, we established BT474 trastuzumab-sensitive (TRZ_S) and trastuzumab-resistant (TRZ_R) cells to investigate the association between PDGFR signaling and TRZ resistance. We found that PDGFRA was significantly upregulated in the BT474 TRZ_R cells. In addition, IL-6 expression, which was also found to be upregulated in the TRZ_R cells, was induced by PDGFC, a ligand of PDGFR. Next, we investigated the effects of ponatinib and sunitinib, PDGFR inhibitors, on the BT474 TRZ_R and HCC1954 (TRZ-resistant cell line) cells. These inhibitors decreased cell viability and migration in a dose-dependent manner. Additionally, IL-6 expression was decreased by ponatinib in both the BT474 TRZ_R and HCC1954 cells. In contrast, IL-6 was not suppressed by TRZ, implying that the PDGFRA/STAT3/IL-6 axis is associated with resistance to TRZ. In addition, we found that STAT3 and ERK phosphorylation were increased in the BT474 TRZ_R cells. IL-6 expression was suppressed by a STAT3 inhibitor, indicating that IL-6 expression is modulated downstream of STAT3. Taken together, these results suggest that PDGFRA could serve as a therapeutic target to overcome TRZ resistance.

## 1. Introduction

Among all of the breast cancer subtypes, about 15% of breast cancers are positive for HER2 (HER2+) [1,2]. The characteristics of HER2+ breast cancers are that the cancer cells are vigorous in growth and metastasis, ultimately leading to poor prognosis with a decreased overall survival rate of the patients [1,2]. For HER2+ breast cancer patients, targeting HER2 using anti-HER2 monoclonal antibody drugs such as trastuzumab (TRZ, also known as herceptin) has been an attractive therapeutic approach [3]. However, about 25% of HER2+ breast cancers do not initially respond to TRZ [4], and 70% of TRZ-responsive metastatic cancers progress despite treatment within the first year due to the acquisition of resistance to TRZ [5]. For other treatment options, drug-conjugated HER2-targeting monoclonal antibodies or tyrosine kinase inhibitors have been developed. Although clinical efficacy of targeting HER2 has been improved by the development and use of various HER2-targeted drugs, de novo and acquired resistance remains a major therapeutic challenge.

Platelet-derived growth factor (PDGF) has been previously identified as a critical regulator of cell proliferation, migration, and angiogenesis in various type of cancers [6]. The PDGF family consists of five isoforms (PDGF-AA, -BB, -AB, -CC, and -DD). Binding of dimeric PDGF to their receptors such as PDGF receptor alpha (PDGFRA) and PDGF receptor beta (PDGFRB) can result in activation of the downstream signaling pathway [7]. These receptors have different affinities for ligands. PDGFRA has a high affinity for PDGFA, PDGFB, and PDGFC, whereas PDGFRB has a high affinity for PDGFB and PDGFD [8,9]. Through immunohistochemical analysis of invasive ductal breast carcinoma tissues, PDGFRA was associated with lymph node metastasis and HER2 expression [10]. When PDGFRA, PDGFRB, and PDGF-CC protein expression levels were observed in the breast tumor samples, PDGF-CC expression was found to be correlated with poor prognosis and PDGFRA was associated with lymph node metastasis and recurrence [11]. Additionally, high stromal PDGFRB expression was associated with significantly shorter recurrence-free and breast cancer-specific survival [12]. PDGFC is expressed in tumor cells including glioblastoma and lung carcinoma [13,14]. The PDGFC/PDGFR complex can trigger downstream signal transduction pathways including extracellular signal-regulated kinase 1/2 (ERK) and phosphatidylinositol 3-kinase (PI3K)/AKT, inducing tumor progression, angiogenesis, and survival in a variety of tumor cells [15,16]. Although previous studies have indicated the signaling pathway of PDGFs and their receptor binding complexes in cancer, the association between PDGF signaling and TRZ resistant breast cancer still remains elusive.

The aim of the present study was to investigate the role of PDGFRA in HER2+ breast cancer. We generated TRZ_R cells and found that PDGFRA expression was increased in the TRZ_R cells compared to the TRZ_S cells. Therefore, we explored the association between PDGFRA and TRZ-resistance. We analyzed the difference in survival rates according to the expression of PDGFRA in HER2+ breast cancer patients and further investigated the efficacy of PDGFR inhibitors such as ponatinib or sunitinib for the treatment of TRZ-resistant breast cancer. We also explored the downstream signaling pathways of the PDGFC and PDGFRA complex in TRZ resistant breast cancer cells.

## 2. Materials and Methods

### 2.1. Cell Culture and Trastuzumab (TRZ)-Resistant Cell Establishment

SKBR3 (TRZ-sensitive cell line), HCC1954 (TRZ-resistant cell line), and BT474 cells were cultured in RPMI medium 1640 (Life Technologies, Rockville, MD, USA) supplemented with 10% fetal bovine serum (FBS; Hyclone, Logan, UT, USA), 2 mM glutamine, 100 IU/mL penicillin, and 100 μg/mL streptomycin. All cell lines were maintained at 37 °C under a humidified incubator with 5% CO_2_ [17,18].

TRZ-resistant cells (TRZ_R) were established by culturing epithelial BT474 breast cancer cells (TRZ-sensitive cells, TRZ_S) in the presence of 50 μg TRZ for around six months [19]. Parental cells (TRZ_S) were cultured in parallel to resistant ones without adding TRZ.

### 2.2. Treatment of Cell Lines

For the treatment of TRZ, each cell was seeded on 6-well plates and after 24 h, TRZ was treated to the cells at 50 μg/mL for 48 h. The cell cycle was analyzed by flow cytometry. For the treatment of ponatinib and sunitinib, each cell (2 × 10^3^ cells/well) was seeded on 96-well plates and after 24 h, ponatinib (Selleck Chemicals, Houston, TX, USA), sunitinib (Selleck Chemicals), or S3I-201 were treated at the indicated concentration for 48 h and for 10 days. In the cell invasion assay, each cell was seeded in the Boyden-chamber inserts with or without 1 μM pontatinib or sunitinib for 24 h. Phosphate-buffered saline was treated as a vehicle control for TRZ treatment and dimethyl sulfoxide was treated as a vehicle control for inhibitors including ponatinib, sunitinib, and S3I-201.

### 2.3. Clinical Implication of PDGFRA in HER2+ Breast Cancers

Using the Kaplan–Meier plotter database, the alteration in relapse-free survival (RFS: low group = 305, high group = 577) was analyzed according to the PDGFRA mRNA expression in HER2+ breast cancer patients [20]. The probe set that was used for the analysis was 203131_at. The patients were split by using the ‘Auto select best cutoff’ option. To restrict the patients to HER2+ breast cancer patients, the ‘HER2 status—array’ was set to ‘HER2-positive’. Additionally, to restrict the breast cancer patients to ER- HER2+ breast cancer patients, ‘ER status—IHC’ was set to ‘ER negative’ and ‘HER2 status—array’ was set to ‘HER2-positive’. To restrict the breast cancer patients to ER+ HER2+ breast cancer patients, the ‘ER status—IHC’ was set to ‘ER positive’ and ‘HER2 status—array’ was set to ‘HER2-positive’. Other options were unaltered. The hazard ratio (HR) and 95% confidence intervals as well as the log rank P were calculated and displayed on the webpage (https://kmplot.com/analysis/index.php?p=service&cancer=breast, accessed on 20 January 2023). A *p*-value < 0.05 was considered to be statistically significant.

### 2.4. RNA Extraction and Reverse Transcription 

The total RNA was isolated using TRI Reagent (Molecular Research Center, Inc., Cincinnati, OH, USA) following the manufacturer’s instructions. One microgram of RNA was used for cDNA synthesis with a first-strand cDNA synthesis kit (MBI Fermentas, Hanover, MD, USA) following the manufacturer’s protocol in a 20 µL reaction volume. 

### 2.5. Quantitative Real-Time PCR (qPCR)

Each gene expression was quantified by real-time PCR using a SensiMix SYBR Kit (Bioline Ltd., London, UK) on an ABI PRISM 7900HT instrument (Applied Biosystems, CA, USA). Sequences of the primer sets used in this study were as follows. IL-6: forward, 5′-AAT TCG GTA CAT CCT CGA CGG-3′ and reverse, 5′-GGT TGT TTT CTG CCA GTG CC-3′; Glyceraldehyde 3-phosphate dehydrogenase (GAPDH, used as an endogenous control): forward, 5′-ATT GTT GCC ATC AAT GAC CC-3′ and reverse, 5′-AGT AGA GGC AGG GAT GT-3′. All reactions were performed in triplicate. Data were analyzed and normalized to the expression of the housekeeping gene *GAPDH* as an internal control. Fold change of the target gene was evaluated by the comparative CT method (2^−∆∆CT^).

### 2.6. Western Blots

Whole cell lysates were harvested using the PRO-PREPTM Protein Extraction Solution (iNtRON, Sungnam, Korea). Western blot analysis was performed as described previously [21,22]. Blots were incubated with anti-PDGFRA (sc-398206, 1:1000, Santa Cruz Biotechnology, Inc., Santa Cruz, CA, USA), PDGFRB (sc-374573, 1:1000, Santa Cruz), p-STAT3 (ab76315, 1:100000, Abcam), p-ERK (sc-7383, 1:1000, Santa Cruz), t-STAT3 (sc-8019, 1:1000, Santa Cruz), t-ERK (#9102, 1:1000, Cell signaling technology, Danvers, MA, USA), t-HER2 (sc-33684, 1:5000, Santa Cruz), or β-actin (LF-PA0207, 1:5000, AbFrontier Co. Ltd., Seoul) antibodies in 1% Tris-buffered saline with 0.01% Tween-20 (TBST) at 4 °C overnight (O/N). Blots were washed 3–4 times in TBST and incubated with appropriate secondary antibodies in TBST buffer for 1 h at room temperature. Blots were washed 3–4 times with TBST buffer and visualized with the ECL™ Western Blotting Detection Reagent (GE Healthcare, Chicago, IL, USA). 

### 2.7. Enzyme-Linked Immunosorbent Assays (ELISAs)

Secreted IL-6 in conditioned culture medium was measured using ELISA kits (KomaBiotech, Seoul, Korea) following the manufacturer’s instructions. Briefly, SKBR3, HCC1954, and BT474 TRZ_S and TRZ_R breast cancer cells were seeded into 6-well plates and then cultured with fresh serum-free medium. After 48 h, conditioned culture media were harvested to quantify the soluble IL-6 protein according to the manufacturer’s protocol using a microplate reader (Spectra max 190, Molecular Devices, Sunnyvale, CA, USA).

### 2.8. Boyden Chamber Assay

Cell migration was analyzed using a Boyden chamber assay as described previously [23]. HCC1954 or BT474 TRZ_R breast cancer cells were placed in the upper compartment of the Boyden-chamber inserts with an 8-μm pore size fitted in 24-well plates (Becton-Dickinson, San Diego, CA, USA). Fresh culture medium with 10% FBS was added to the lower compartment of the chamber. At the end of the assay, after the cells on the upper side of the filter were removed, the filter was fixed with 100% methanol, and then stained with 0.1% crystal violet solution. The migrated cells were analyzed using a Scanscope XT scanner (Aperio Technologies, Vista, CA, USA).

### 2.9. Colony Formation Assays

For the colony formation assays, BT474 TRZ_R and HCC1954 breast cancer cells were plated into 6-well plates (2 × 10^3^ cells/well) and incubated at 37 °C, O/N. The next day, the cells were treated with 2 μM ponatinib or sunitinib (Selleck Chemicals) and then incubated for an additional 10 days. Cell colonies were fixed with 10% ethanol, stained with 0.01% crystal violet, and observed using a CK40 inverted microscope (Olympus, Tokyo, Japan). 

### 2.10. MTT Assay

BT474 TRZ_R and HCC1954 breast cancer cells were plated for 3-(4,5-dimethylthiazol-2-yl)-2,5-diphenyltetrazolium bromide (MTT) assays in 96-well plates at a density of 2 × 10^3^ cells per well and cultured in 100 μL medium per well at 37 °C. The next day, cells were treated with ponatinib or sunitinib at the indicated concentrations for 48 h. Cell proliferation was determined by adding 10 μL of 5 mg/mL MTT (Sigma, St. Louis, MO, USA) to each well. The optical density was read at a wavelength of 590 nm using a microplate reader (Spectra max 190, Molecular Devices, Sunnyvale, CA, USA). 

### 2.11. Statistical Analysis

The data were analyzed and graphs were generated using Microsoft Excel 2016 (Microsoft, Redmond, WA, USA) and GraphPad Prism 8 software (La Jolla, CA, USA). Results are presented as the mean ± standard error of the mean (S.E.M.). For the data analysis, *p* values were calculated using one-way analysis of variance (ANOVA) or the Student’s t-test (unpaired, two-tailed). Statistical significance was considered at a *p*-value < 0.05. All experiments were repeated at least three times independently.

## 3. Results

### 3.1. High Expression of PDGFRA Predicts Poor Survival of Patients with HER2+ Breast Cancers

As shown in Figure 1A, we established BT474 trastuzumab-sensitive (TRZ_S) and trastuzumab-resistant (TRZ_R) breast cancer cell models to find therapeutic targets of TRZ resistance. Sensitivity to TRZ in the established cell models were verified by cell cycle assays. G0/G1 phase arrest of TRZ_S cells was markedly induced by TRZ compared to the TRZ_R cells (Figure 1B). Since high stromal PDGFRB expression is known to be correlated with shorter survival in breast cancer, we checked the expression levels of PDGFRA and PDGFRB in TRZ_S and TRZ_R derived from the parental BT474 cells. Interestingly, we found that PDGFRA expression, but not PDGFRB expression, was significantly increased in the TRZ_R cells compared to TRZ_S cells (Figure 1C). Furthermore, we analyzed the clinical implication of PDGFRA expression in HER2+ breast cancers. The results showed that HER2+ breast cancer patients with high PDGFRA expression had worse relapse-free survival (RFS, *p* = 0.02) than those with low PDGFRA expression (Figure 1D). In addition, when the survival rates were compared by further stratifying the HER2+ patients with ER status, patients with high PDGFRA expression levels had poorer RFS in both the ER− HER2+ and ER+ HER2+ patients (Appendix A). These results imply that the PDGFRA expression level might have a direct or indirect association with HER2+ breast cancer progression.

### 3.2. IL-6 Expression Level Is Upregulated by PDGFC Treatment in HER2+ Breast Cancer Cells

Since IL-6 is known to be associated with TRZ resistance in HER2+ breast cancer, we sought to analyze the expression levels of IL-6 in TRZ_S and TRZ_R derived from the parental BT474 cells. The results showed that the levels of IL-6 mRNA (Figure 2A) and protein (Figure 2B) were significantly higher in the TRZ_R cells compared to the TRZ_S cells. As shown in Appendix A, when the IL-6 levels were compared between the SKBR3 and HCC1954 cells, representative breast cancer cell lines that are sensitive and resistant to TRZ, respectively [24], IL-6 expression was significantly higher in the HCC1954 cells compared to those of the SKBR3 cells. However, the levels of IL-6 mRNA and protein expression were not affected by TRZ treatment in the BT474 TRZ_R and HCC1954 cells (Appendix A). A previous study demonstrated that PDGF treatment led to the induction of genes in the IL-6 pathway using mouse embryonic fibroblasts [25]. Therefore, we next tried to investigate whether there was an association between PDGF signaling and IL-6 in HER2+ breast cancer through observing the alteration of IL-6 expression by PDGFR activation. As expected, PDGFC, one of the PDGFR ligands, upregulated IL-6 expression in the SKBR3 cells (Appendix A). Taken together, these results imply that the PDGFC/PDGFR signaling axis might serve as a novel mechanism involved in TRZ resistance through inducing IL-6, which is a proinflammatory cytokine.

### 3.3. Pharmacological Effects PDGFR Inhibitors on TRZ-Resistant Breast Cancer Cells

As an attempt to block the PDGFC/PDGFR signaling axis, we investigated the pharmacological effects of two PDGFR inhibitors (Figure 3A) on the growth and migration of BT474 TRZ_R and HCC1954 cells. We investigated the effects of two PDGFR inhibitors on cell viability by treating the cells with ponatinib or sunitinib at the indicated concentrations for 48 h. As shown in Figure 3B, cell viability was decreased by both ponatinib and sunitinib treatment. Ponatinib at 2 µM decreased the viability of BT474 TRZ_R and HCC1954 cells to 45.3 ± 1.5% and 52.4 ± 3.4% (compared to the control), respectively (Figure 3B). Sunitinib at 2 µM decreased the viability of BT474 TRZ_R and HCC1954 cells to 69.8 ± 1.0% and 84.0 ± 1.9% (compared to control), respectively (Figure 3B). In particular, the pharmacological effect of ponatinib was better than that of sunitinib, showing a dose-dependent effect on the cell viability of both cells (Figure 3B). 

We also examined the effects of two PDGFR inhibitors on cell proliferation using colony formation assays. Both ponatinib and sunitinib inhibited the growth of the BT474 TRZ_R and HCC1954 cells (Figure 3C). Ponatinib and sunitinib at 1 µM decreased the colony numbers of BT474 TRZ_R cells to 12.4 ± 2.0% and 30.6 ± 3.7% (compared to the control), respectively, and HCC1954 cells to 29.8 ± 2.6% and 52.2 ± 4.1% (compared to the control), respectively (Figure 3C). In addition, Transwell assays revealed that sunitinib and ponatinib at 1 µM significantly inhibited the migration capacity of the BT474 TRZ_R cells to 38.2 ± 3.1% and 16.4 ± 1.9% (compared to the control), respectively, and HCC1954 cells to 73.6 ± 3.8% and 12.4 ± 2.0% (compared to control), respectively (Figure 3D). These findings suggest that PDGFR inhibitors such as ponatinib and sunitinib can inhibit the proliferation and migration of TRZ-resistant cells. Additionally, these results collectively show that the pharmacological effects of ponatinib are superior to that of sunitinib when treated to the TRZ-resistant cells.

### 3.4. Ponatinib Inhibits PDGFC/PDGFR/STAT3 Signaling Pathway in TRZ-Resistant Breast Cancer Cells

To explore the molecular basis of the anti-proliferative effects of ponatinib on TRZ-resistant breast cancer cells, human phosphor-kinase array kits were employed to detect the phosphorylation levels of a panel of phospho-kinases in the whole cell lysates of the BT474 TRZ_S and TRZ_R cells. We found that the levels of signal transducers and activators of transcription 3 (STAT3) and ERK phosphorylation were significantly increased in the TRZ_R cells compared to the TRZ_S cells (Figure 4A). 

Next, we analyzed the effect of ponatinib on STAT3 and ERK phosphorylation in the BT474 TRZ_R and HCC1954 cells. After the BT474 TRZ_R and HCC1954 cells were treated with 2 μM ponatinib for 1 h, the levels of STAT3 phosphorylation, but not ERK phosphorylation, were reduced by ponatinib in both the BT474 TRZ_R and HCC1954 cells (Figure 4B).

Furthermore, when we treated the BT474 TRZ_R and HCC1954 cells with 2 μM ponatinib for 48 h under serum-free conditions, we found that IL-6 expression was downregulated by ponatinib treatment (Figure 4C,D). Ponatinib at 2 μM decreased the secreted IL-6 levels in the BT474 TRZ_R cells to 2.44 ± 0.02 pg/mL compared to the control (7.11 ± 0.84 pg/mL) (Figure 4C). Under the same conditions, the levels of IL-6 secretion in HCC1954 cells were also inhibited to 22.31 ± 5.12 pg/mL compared to the control (60.60 ± 3.92 pg/mL) (Figure 4D). Finally, we examined the effect of S3I-201, an inhibitor of STAT3, on IL-6 expression through treating the cells with 20 μM S3I-201 for 48 h. The levels of IL-6 secretion in both the BT474 TRZ_R and HCC1954 cells were decreased by treatment with S3I-201 (Figure 5A,B), indicating that IL-6 was regulated downstream of STAT3.

## 4. Discussion

To date, the clinical implication of PDGFRA expression in HER2+ breast cancers has not been fully elucidated yet. PDGFs and their receptors (PDGFRA and PDGFRB) are expressed in a variety of malignant tumor cells and tissues such as breast cancer and neuroendocrine tumors [18,26,27]. PDGFC expression is ubiquitous in brain tumors. It is known that PDGFC can trigger the initiation and progression of brain tumors [13]. High PDGFRA and PDGFC expression can promote tumor proliferation, angiogenesis, migration, and invasion [11,18,26]. Consistent with these reports, we also observed that HER2+ breast cancer patients with low PDGFRA expression had better RFS than those with high PDGFRA expression. PDGFRA expression, but not PDGFRB expression, was significantly increased in the BT474 TRZ_R cells compared to the TRZ_S cells. These results demonstrate that high PDGFRA expression is associated with decreased survival and TRZ resistance in HER2+ breast cancer. 

IL-6 is produced and secreted in inflammatory breast cancer. It is known that IL-6 can promote breast cancer progression through increasing cell invasion, angiogenesis, and metastasis [28,29,30]. Induction of IL-6 can increase the tumor stage with lymph node involvement, recurrence risk, and distant metastasis in breast cancers [31]. Previous studies have shown that IL-6 plays an essential role in regulating the self-renewal of stem cells and TRZ resistance in HER2+ breast cancer [32,33]. In addition, autocrine production of IL-6 can induce multidrug resistance [34]. Knocking down PTEN expression in HER2+ breast cancer cells induces trastuzumab resistance through the activation of an IL-6 inflammatory feedback loop [33]. Zhong et al. reported that MEDI5117, which is a novel high-affinity anti-IL-6 antibody, completely inhibits IL-6-induced activation of STAT3 and then suppresses the tumor angiogenesis as well as the growth of several tumor types including prostate cancer [35]. Consistent with these reports, we also found that the level of IL-6 expression was significantly increased in the BT474 TRZ_R cells when compared to the TRZ_S cells. Interestingly, elevated IL-6 expression was decreased by ponatinib, but not by TRZ. Since TRZ is known to preferentially target HER2 homodimers rather than heterodimers [36], and BT474 expresses other EGFR family members including EGFR and HER3, we assumed that HER2 heterodimer signaling, rather than HER2 homodimer signaling, is associated with the increased PDGFRA levels of the TRZ_R cells. These results demonstrate that PDGFC/PDGFRA signaling induces IL-6 expression and implies that the PDGFR/IL-6 axis is associated with TRZ resistance.

The PDGF and PDGF receptor signaling pathway can activate downstream signaling molecules such as the Janus kinase (JAK)/STAT pathway and mitogen-activated protein kinase (MAPK)/ERK pathway [37,38]. The combination of melatonin and sorafenib synergistically suppresses tumor growth in pancreatic cancer xenograft models through the downregulation of the PDGFR-β/STAT3 signaling pathway [39]. Huang et al. reported that STAT3 could potentially upregulate the autocrine production of IL-6 in cancer cells to enhance cancer progression and drug resistance [40]. In contrast, inhibiting STAT3 by specific inhibitors or STAT3 siRNA decreases the levels of IL-6, IL-10, and VEGF mRNA expression in melanoma cells [41]. Here, our results showed that the levels of STAT3 and ERK phosphorylation were dramatically increased in the BT474 TRZ_R cells compared to the TRZ_S cells. Therefore, we evaluated the effects of ponatinib, a PDGFR inhibitor, on the downstream signals of the BT474 TRZ_R cells. Ponatinib inhibited the phosphorylation of STAT3, but not that of ERK. These results show that PDGFR signaling activates STAT3 in TRZ-resistant breast cancer. 

In conclusion, we evaluated the clinical implication of PDGFRA expression and the pharmacological effects of PDGFR inhibitors on TRZ-resistant cells. PDGFRA expression is upregulated in TRZ-resistant cells and inversely correlated with HER2+ breast cancer patient survival. Although our results show that ponatinib (PDGFRA inhibitor) is more effective than sunitinib (PDGFRB inhibitor) in reducing growth and migration of the cancer cells, both inhibitors were capable of suppressing cell proliferation and migration. IL-6, an inflammatory cytokine, is one of the TRZ resistance-related genes. Ponatinib treatment downregulated IL-6 expression, but TRZ failed to decrease IL-6. Taken together, these results demonstrate that abnormal PDGFRA expression is associated with resistance to TRZ through activating STAT3, which leads to the upregulation of IL-6. Therefore, targeting the PDGFRA/STAT3/IL-6 pathway using PDGFRA inhibitors might serve as a therapeutic option to reduce tumor aggressiveness in HER2+ breast cancer. 

## Figures and Tables

**Figure 1 biomedicines-11-00675-f001:**
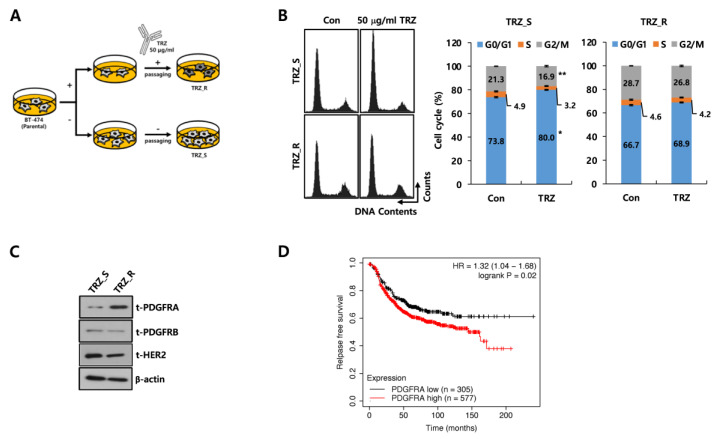
The high expression of the platelet-derived growth factor receptor alpha (PDGFRA) predicts the poor survival of patients with HER2-positive (HER2+) breast cancer. (**A**) A schematic model showing the generation of the trastuzumab-resistant (TRZ_R) cell line. BT474 TRZ_R cells were established by culturing wild type BT474 breast cancer cells in the presence of 50 μg/mL TRZ for around 6 months. (**B**) BT474 trastuzumab-sensitive (TRZ_S) and TRZ_R cells were treated with or without 50 μg/mL TRZ for 48 h. Then, the cell cycle was analyzed. Each experiment was carried out three independent times. Data are presented as the mean ± SEM (* *p* < 0.05, ** *p* < 0.01). (**C**) PDGFRA, PDGFR beta (PDGFRB), HER2, and β-actin levels were analyzed by Western blotting. (**D**) Relapse-free survival (RFS) of the HER2+ breast cancer patients with high (n = 577) or low (n = 305) PDGFRA expression were analyzed using the Kaplan–Meier plotter database. The patients were split by using the Auto select best cutoff option.

**Figure 2 biomedicines-11-00675-f002:**
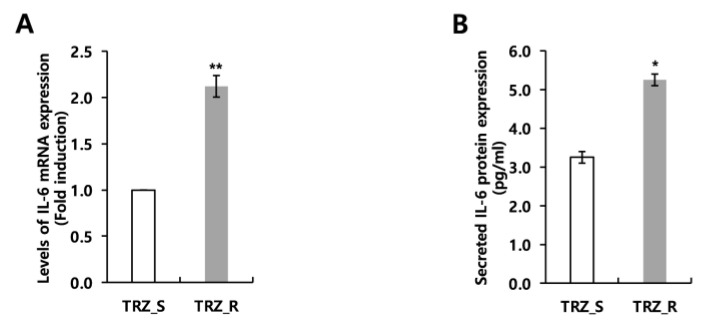
IL-6 expression is upregulated in TRZ_R cells compared to the TRZ_S cells. The levels of IL-6 mRNA and protein expression were analyzed by quantitative real-time PCR (qPCR) (**A**) and enzyme-linked immunosorbent assay (ELISA) (**B**), respectively. Each experiment was carried out three independent times. Data are presented as the mean ± SEM (* *p* < 0.05, ** *p* < 0.01).

**Figure 3 biomedicines-11-00675-f003:**
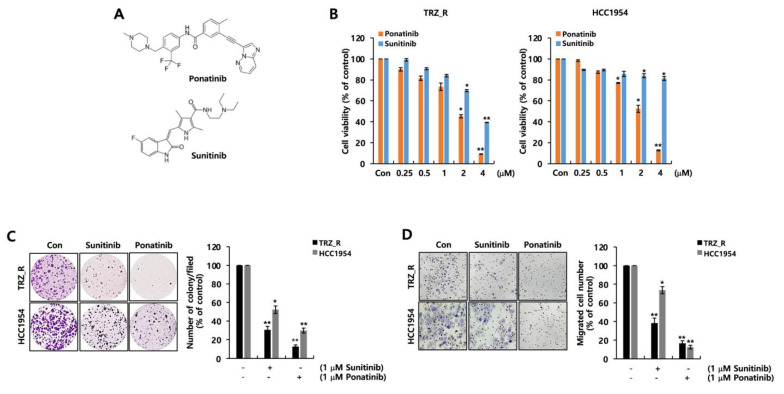
Pharmacological effects of two PDGFR inhibitors on the TRZ-resistant breast cancer cells. (**A**) Chemical structures of ponatinib and sunitinib. (**B**) Cells were seeded into 96-well plates and treated with ponatinib or sunitinib at the indicated concentrations for 48 h. Cell viability was analyzed by the 3-(4,5-dimethylthiazol-2-yl)-2,5-diphenyltetrazolium bromide (MTT) assay. (**C**) Cell proliferation was analyzed by the colony formation assay. (**D**) Cell migration capacities were analyzed using the Boyden chamber assays. Each experiment was carried out three independent times. Data are presented as the mean ± SEM (* *p* < 0.05, ** *p* < 0.01).

**Figure 4 biomedicines-11-00675-f004:**
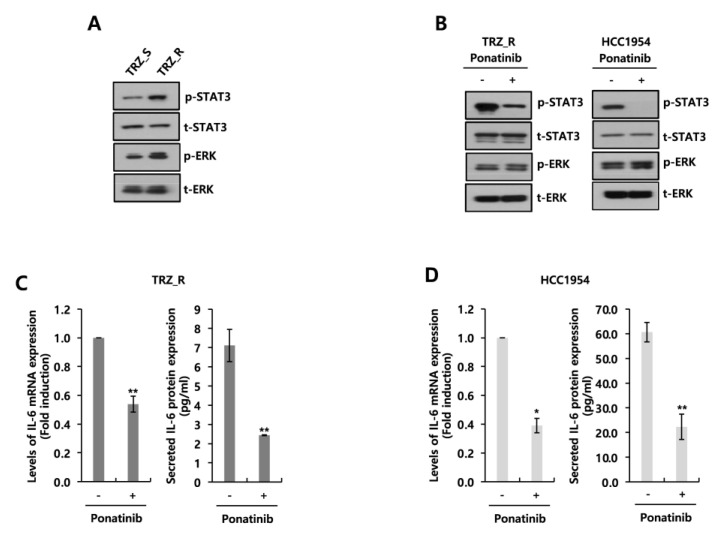
Ponatinib inhibits the PDGFC/PDGFR/STAT3 signaling pathway in the TRZ-resistant breast cancer cells. BT474 TRZ_R and HCC1954 cells were treated with 2 μM ponatinib for 48 h. (**A**,**B**) Levels of t-STAT3, p-STAT3, and ERK protein in whole cell lysates were analyzed by Western blotting. (**C**,**D**) Levels of IL-6 mRNA (in whole cell lysates) and protein (in conditioned culture media) were analyzed by qPCR and ELISA, respectively. Each experiment was carried out three independent times. Data are presented as the mean ± SEM (* *p* < 0.05, ** *p* < 0.01).

**Figure 5 biomedicines-11-00675-f005:**
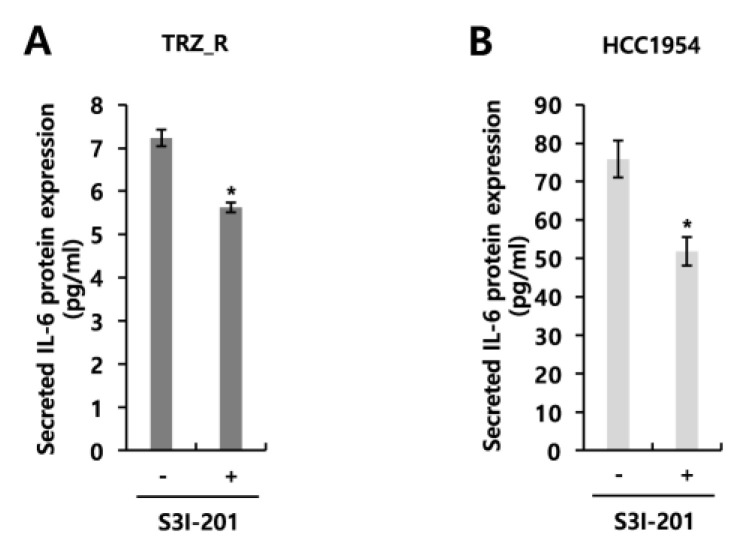
S3I-201 (STAT3 inhibitor) inhibits IL-6 expression in TRZ-resistant breast cancer cells. BT474 TRZ_R and HCC1954 cells were treated with 20 μM S3I-201 for 48 h. (**A**,**B**) Levels of IL-6 protein (in conditioned culture media) were analyzed by ELISA. Data are presented as the mean ± SEM from triplicate measurements (* *p* < 0.05).

## Data Availability

The datasets used and/or analyzed during the current study are available from the corresponding author on reasonable request.

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
