# Peer review of "Suppression of Platelet-Derived Growth Factor Receptor-Alpha Overcomes Resistance to Trastuzumab through STAT3-Dependent IL-6 Reduction in HER2-Positive Breast Cancer Cells"

_biomedicines, 2023, doi:10.3390/biomedicines11030675_

Round 1

Reviewer 1 Report

Kim et al report the clinical significance of platelet-derived- growth factor alpha (PDGFA) in HER2+ breast cancer and the therapeutic potential of PDGFR inhibitors in overcoming trastuzumab (TRZ) resistance. They demonstrate increased PDGFRA expression is associated with poorer survival, and that PDGFRA expression was significantly increased in the TRZ resistant breast cancer cell line which they established from BT474.

They show that IL-6 expression was increased in TRZ resistant cell lines, which could also be upregulated by treatment with PDGFC. IL-6 however was not affected by TRZ treatment. Treatment by PDGFR inhibitors, Sunitinib and Ponatinib, decreased the viability and inhibited the migration capacity of TRZ resistant cell lines. TRZ resistant cells showed higher expression of phosphorylated STAT3 and phosphorylated ERK compared with parenteral TRZ sensitive cell lines. Treatment with Ponatinib resulted in downregulation of IL-6 expression, and suppression of pSTAT3, but not pERK. Treatment with STAT3 inhibitor resulted in decreased IL-6 expression, indicating IL-6 is regulated downstream of STAT3.  The authors conclude these results show that PDGFRA signalling activates STAT3 in TRZ resistant breast cancer, and that targeting the PDGFRA/STAT3/IL-6 pathway using PDGRFA inhibitors might serve as a therapeutic option to overcome TRZ resistance.

There are several major concerns in this manuscript which must be addressed before the conclusions can be accepted.

1. With regard to the clinical implication of PDGFRA in HER2+ breast cancers, there is no indication of the database source from which expression levels were obtained, nor the number of cases involved. Also, how were the high expression and low expression cases dichotomized? Was the cut of taken at median value, or quartile?

2. The statement “PDGFRA signalling activates STAT3 in TRZ resistant breast cancer” makes the assumption that PDGFC treatment, which upregulates IL-6 expression, indeed activates the PDGFRA signalling pathway. The authors should at least demonstrate that PDGFC treatment would upregulate PDGFRA expression.

3. The final statement “targeting the PDGFRA/STAT3/IL-6 pathway using PDGRFA inhibitors might serve as a therapeutic option to overcome TRZ resistance” is also overstated. While treatment by PDGFR inhibitors decreased viability and inhibited the migration capacity of TRZ resistant cell lines, in these experiments the cell lines were not being treated with TRZ. Hence no conclusion can be made about TRZ resistance status. As reported in reference 28 (Jansson et al), the PDGF pathway in breast cancer is linked to TNBC subtype and tumor aggressiveness. Hence the effect of targeting this pathway could reduce tumor aggressiveness, as was demonstrated.

4. The authors should at least attempt to explain why IL-6 however was not affected by TRZ treatment.

5. Regarding the TRZ resistant cell line established from BT474 TRZ sensitive cell line, it would be desirable to show it retained it HER2 overexpression.

  Minor comments

1. Reference 18 and 29 report IL-8, not IL-6

2.  Line 211 – 214 “ transwell assays revealed that ponatinib and sunitinib at 1 μM significantly inhibited the migration capacity of BT474 TRZ_R cells…..” The data shown in Fig. 3D indicate it should be sunitinib and ponatinib instead.

Author Response

  1. With regard to the clinical implication of PDGFRA in HER2+ breast cancers, there is no indication of the database source from which expression levels were obtained, nor the number of cases involved. Also, how were the high expression and low expression cases dichotomized? Was the cut of taken at median value, or quartile?

We thank the reviewer’s comment. Previously, we have stated the explanation for the patient survival data regarding PDGFRA expression in the Materials and Methods section as following: “Using the Kaplan-Meier plotter database, the alteration in relapse-free survival (RFS: low group = 180, high group= 72) and distant metastasis-free survival (DMFS: low group = 90, high group = 36) was analyzed according to PDGFRA mRNA expression in HER2+ breast cancer patients [17]. Hazard ratio (HR) and 95% confidence intervals as well as log rank P were calculated and displayed on the webpage (https://kmplot.com/analysis/index.php?p=service&cancer=breast). P-value < 0.05 was considered to be statistically significant.”. The patients were split by using the ‘Auto select best cutoff’ option in the KM plotter database.

However, the results were obtained more than years ago, and therefore we have updated the survival plots. We have included the detailed description of how the data was obtained in the Materials and Methods section and the Figure Legends.

  1. The statement “PDGFRA signalling activates STAT3 in TRZ resistant breast cancer” makes the assumption that PDGFC treatment, which upregulates IL-6 expression, indeed activates the PDGFRA signalling pathway. The authors should at least demonstrate that PDGFC treatment would upregulate PDGFRA expression.

We appreciate the reviewer’s comment. We have made the assumption that PDGFC would activate PDGFRA since PDGFC and PDGFA are known as principal ligands for PDGFRA through previous studies (PMID: 26988758). However, since our results are insufficient to directly state that “PDGFRA signaling activates STAT3 in TRZ-resistant breast cancer”, we have changed the statement as following: “These results show that PDGFR signaling activates STAT3 in TRZ-resistant breast cancer”.

  1. The final statement “targeting the PDGFRA/STAT3/IL-6 pathway using PDGRFA inhibitors might serve as a therapeutic option to overcome TRZ resistance” is also overstated. While treatment by PDGFR inhibitors decreased viability and inhibited the migration capacity of TRZ resistant cell lines, in these experiments the cell lines were not being treated with TRZ. Hence no conclusion can be made about TRZ resistance status. As reported in reference 28 (Jansson et al), the PDGF pathway in breast cancer is linked to TNBC subtype and tumor aggressiveness. Hence the effect of targeting this pathway could reduce tumor aggressiveness, as was demonstrated.

We thank the reviewer’s careful reading. We agree to the reviewer’s comment in that stating PDGFRA inhibition could serve as a therapeutic target to overcome trastuzumab resistance is overstating given our current data. Therefore, we have modified the final statement as following: “Therefore, targeting the PDGFRA/STAT3/IL-6 pathway using PDGFRA inhibitors might serve as a therapeutic option to reduce tumor aggressiveness in HER2+ breast cancer.”

  1. The authors should at least attempt to explain why IL-6 however was not affected by TRZ treatment.

We appreciate the reviewer’s comment. We have included the following sentence in the discussion section to explain why IL-6 expression was not affected by trastuzumab treatment: “Since TRZ is known to preferentially target HER2 homodimers rather than heterodimers (12), and BT474 expresses other EGFR family members including EGFR and HER3, we assume that HER2 heterodimer signaling, rather than HER2 homodimer signaling, is associated with the increased PDGFRA levels of the TRZ_R cells.”

  1. Regarding the TRZ resistant cell line established from BT474 TRZ sensitive cell line, it would be desirable to show it retained it HER2 overexpression.

We thank the reviewer’s comment. We have included the HER2 expression data of BT474 trastuzumab-sensitive cells in Figure 1 panel C.

  Minor comments

  1. Reference 18 and 29 report IL-8, not IL-6

We thank the reviewer’s careful reading. We have appropriately changed the references for the sentence “It is known that IL-6 can promote cancer progression through increasing cell invasion, angiogenesis, and metastasis in HER2+ and triple-negative breast cancer” in the discussion section.

  1. Line 211 – 214 “ transwell assays revealed that ponatinib and sunitinib at 1 μM significantly inhibited the migration capacity of BT474 TRZ_R cells…..” The data shown in Fig. 3D indicate it should be sunitinib and ponatinib instead.

We thank the reviewer’s careful reading. We have modified the sentence as following: “In addition, transwell assays revealed that sunitinib and ponatinib at 1 µM significantly inhibited the migration capacity of BT474 TRZ_R cells to 38.2 ± 3.1% and 16.4 ± 1.9% (compared to control), respectively; HCC1954 cells to 73.6 ±3.8% and 12.4 ± 2.0% (compared to control), respectively (Fig. 3D).”

Reviewer 2 Report

The authors explored the role of PDGFR in trastuzumab resistance in breast cancer. However, some concerns might be reported about the manuscript.

The introduction section is well written, and the study’s aim is well established.

The authors reported several breast cancer cell lines used in the materials and methods section. However, it is not clear why the authors selected these cell lines. Moreover, the BT474 cell line is a triple-positive cell line (positive for both hormone receptors and HER2+). Please explain why the authors established a trastuzumab (TRZ)- resistance cell line based on this cell line. Why did the authors not use the SKBR3 cell line instead?

The authors should clearly describe all the cell line treatments performed in the manuscript in the materials and methods section.

Concerning the 2.2 section, the authors might describe all the criteria used to generate the survival plots, namely which database was used. The km plot database has more than 1000 cases reported as HER2+, but the authors only presented the analysis considering around 250 patients.

In section 2.5, the authors should report antibody references (if available, the clone used or the catalogue number instead).

In the results section, the authors reported survival analysis in HER2+ breast cancer patients. Survival analysis might be stratified according to hormone receptor status. The estrogen pathway activation also plays an important role in breast cancer development and resistance.

In section 3.2, the authors presented mRNA levels and secreted IL-6 after PDGFC treatment. Data regarding the mRNA levels and secreted IL-6 without any treatment (vehicle condition)  might also be shown in figure 2.

In the following subsection, the authors reported the effect of 2µM concentration on cell viability for both sunitinib and ponatinib. However, for cell proliferation and migration assays, a 1µM concentration was used for both inhibitors. Please explain why.

The authors might be more careful in the discussion section in some sentences. For example, in line 289, “Ponatinib completely inhibited the phosphorylation of STAT3,…”. This is true for the TRZ_R cell line but not for the HCC1954 cell line. The authors might also discuss the differences between analysis on HCC1954 and TRZ_R cell lines.

Minor:

-          A µ is missing in line 125

-          Figure 1B legend might be completed (** reference is missing)

-          Supplementary figures legends are not in accordance with the figures.

Author Response

The authors reported several breast cancer cell lines used in the materials and methods section. However, it is not clear why the authors selected these cell lines. Moreover, the BT474 cell line is a triple-positive cell line (positive for both hormone receptors and HER2+). Please explain why the authors established a trastuzumab (TRZ)- resistance cell line based on this cell line. Why did the authors not use the SKBR3 cell line instead?

We appreciate the reviewer’s kind comments. We have used SKBR3 and HCC1954 cell lines in this study as representative cell lines that are sensitive and resistant to trastuzumab, respectively. We do agree to this comment in that we could have used the SKBR3 cell line to generate a trastuzumab resistant cell line. However, numerous previous studies have utilized the BT474 cell line to generate trastuzumab resistant cell lines (PMID: 17699871, 14750129, 19633047), and therefore we have also chosen the BT474 cell line to generate a trastuzumab resistant cell line.

The authors should clearly describe all the cell line treatments performed in the manuscript in the materials and methods section.

We appreciate the reviewer’s comment. We have described all cell line treatment conditions in the manuscript as a separate section (2.2) in the Materials and Methods section as following:

2.2. Treatment of cell lines    

For treatment of TRZ, each cell was seeded on 6 well plates and after 24 h, TRZ was treated to the cells at 50 μg/ml for 48 h. Cell cycle was analyzed by flow cytometry (Fig. 1B). For treatment of ponatinib and sunitinib, each cell (2 x 103 cells/well) was seeded on 96 well plates and after 24 h, ponatinib, sunitinib, or S3I-201 were treated at the indicated concentration for 48 h (Fig. 3B and Fig. 5A) and for 10 days (Fig. 3C). In cell invasion assay, each cell was seeded in the Boyden-chamber inserts with or without 1 μM pontatinib or sunitinib for 24 h. PBS was treated as a vehicle control for TRZ treatment and DMSO was treated as a vehicle control for inhibitors including ponatinib, sunitinib, and S3I-201.

Concerning the 2.2 section, the authors might describe all the criteria used to generate the survival plots, namely which database was used. The km plot database has more than 1000 cases reported as HER2+, but the authors only presented the analysis considering around 250 patients.

We thank the reviewer’s comment. The current survival plots were generated more than years ago and have not been updated. We have reanalyzed the survival data with HER2+ patients and have changed Figure 1 with the updated survival plots. Also, we have described all the criteria used for generating the plots in the Materials and Methods and the Figure Legends section.

In section 2.5, the authors should report antibody references (if available, the clone used or the catalogue number instead).

We appreciate the reviewer’s comment. We have included the catalogue numbers and dilution ratios for the antibodies in section 2.5.

In the results section, the authors reported survival analysis in HER2+ breast cancer patients. Survival analysis might be stratified according to hormone receptor status. The estrogen pathway activation also plays an important role in breast cancer development and resistance.

We thank the reviewer’s comment. We have also performed the survival analysis of HER2+ breast cancer patients by stratifying the patients according to ER status. We have included the results as Supplementary Figure 1 and mentioned the results in the manuscript as following: “In addition, when survival rates were compared by further stratifying the HER2+ patients with ER status, patients with high PDGFRA expression levels had poorer RFS in both ER- HER2+ and ER+ HER2+ patients (Supplementary Fig. 1).”

In section 3.2, the authors presented mRNA levels and secreted IL-6 after PDGFC treatment. Data regarding the mRNA levels and secreted IL-6 without any treatment (vehicle condition)  might also be shown in figure 2.

We thank the reviewer’s comment. We have mistakenly written the title for Figure 2 as “IL-6 expression is upregulated by PDGFC treatment in HER2+ breast cancer cells”. The experiment in Figure 2 is done without any treatment. We are sorry for causing such confusion and have changed the title for Figure 2 as following: “IL-6 expression is upregulated in TRZ_R cells compared to TRZ_S cells.”

In the following subsection, the authors reported the effect of 2µM concentration on cell viability for both sunitinib and ponatinib. However, for cell proliferation and migration assays, a 1µM concentration was used for both inhibitors. Please explain why.

We appreciate the reviewer’s comment. We have conducted the cell viability assays using increasing concentrations (0.25, 0.5, 1, 2, and 4 μM) but only mentioned the results from 2µM concentrations since that was the lowest concentration which showed significantly reduced cell viability by both ponatinib and sunitinib treatment. The reason we used 1 μM for the following proliferation and migration assays was to show the effects in the lower concentrations, if possible, to emphasize the effects of those inhibitors.

The authors might be more careful in the discussion section in some sentences. For example, in line 289, “Ponatinib completely inhibited the phosphorylation of STAT3,…”. This is true for the TRZ_R cell line but not for the HCC1954 cell line. The authors might also discuss the differences between analysis on HCC1954 and TRZ_R cell lines.

We thank the reviewer’s careful reading. To describe our results more appropriately, we have changed the sentence as following: “After BT474 TRZ_R and HCC1954 cells were treated with 2 mM ponatinib for 1 h, levels of STAT3 phosphorylation, but not ERK phosphorylation, were reduced by ponatinib in both BT474 TRZ_R and HCC1954 cells.”

Minor:

-          A µ is missing in line 125

We thank the reviewer’s careful reading. We have added the missing µ symbol.

-          Figure 1B legend might be completed (** reference is missing)

We thank the reviewer’s careful reading. We have included the explanation for the symbol ** in the figure legends as following: **P < 0.01.

-          Supplementary figures legends are not in accordance with the figures.

We appreciate the reviewer’s comment. We have changed the Supplementary Figure Legends to be in accordance with the figures.

Round 2

Reviewer 2 Report

Survival analysis methods might be more detailed, so could be replicate.
All the other comments, the authors replied clearly

Author Response

Survival analysis methods might be more detailed, so could be replicate.
All the other comments, the authors replied clearly

We appreciate the reviewer's kind comment. We have included the detailed methods that were used for the survival analysis in section 2.3 as following: "

Using the Kaplan-Meier plotter database, the alteration in relapse-free survival (RFS: low group = 305, high group= 577) was analyzed according to PDGFRA mRNA expression in HER2+ breast cancer patients [17]. The probe set that was used for the analysis was 203131_at. The patients were split by using the ‘Auto select best cutoff’ option. For restricting the patients to HER2+ breast cancer patients, ‘HER2 status – array’ was set to ‘HER2-positive’. Also, for restricting the breast cancer patients to ER- HER2+ breast cancer patients, ‘ER status – IHC’ was set to ‘ER negative’ and ‘HER2 status – array’ was set to ‘HER2-positive’; For restricting the breast cancer patients to ER+ HER2+ breast cancer patients, ‘ER status – IHC’ was set to ‘ER positive’ and ‘HER2 status – array’ was set to ‘HER2-positive’. Other options were unaltered. Hazard ratio (HR) and 95% confidence intervals as well as log rank P were calculated and displayed on the webpage (https://kmplot.com/analysis/index.php?p=service&cancer=breast). P-value < 0.05 was considered to be statistically significant."